# Furcellaran Surface Deposition and Its Potential in Biomedical Applications

**DOI:** 10.3390/ijms23137439

**Published:** 2022-07-04

**Authors:** Kateřina Štěpánková, Kadir Ozaltin, Jana Pelková, Hana Pištěková, Ilkay Karakurt, Simona Káčerová, Marian Lehocky, Petr Humpolicek, Alenka Vesel, Miran Mozetic

**Affiliations:** 1Centre of Polymer Systems, University Institute, Tomas Bata University in Zlín, Trida Tomase Bati 5678, 76001 Zlin, Czech Republic; k1_stepankova@utb.cz (K.Š.); ozaltin@utb.cz (K.O.); pistekova@utb.cz (H.P.); ykarakurt@utb.cz (I.K.); s_kacerova@utb.cz (S.K.); humpolicek@utb.cz (P.H.); 2Department of Haematology, Tomas Bata Regional Hospital, Havlickovo Nabrezi 2916, 76001 Zlin, Czech Republic; pelkova@utb.cz; 3Faculty of Humanities, Tomas Bata University in Zlín, Stefanikova 5670, 76001 Zlin, Czech Republic; 4Faculty of Technology, Tomas Bata University in Zlín, Vavreckova 5669, 76001 Zlin, Czech Republic; 5Department of Surface Engineering, Jozef Stefan Institute, Jamova cesta 39, 1000 Ljubljana, Slovenia; alenka.vesel@ijs.si (A.V.); miran.mozetic@ijs.si (M.M.)

**Keywords:** furcellaran, polysaccharide, biopolymer, deposition, cell-surface interaction

## Abstract

Surface coatings of materials by polysaccharide polymers are an acknowledged strategy to modulate interfacial biocompatibility. Polysaccharides from various algal species represent an attractive source of structurally diverse compounds that have found application in the biomedical field. Furcellaran obtained from the red algae *Furcellaria lumbricalis* is a potential candidate for biomedical applications due to its gelation properties and mechanical strength. In the present study, immobilization of furcellaran onto polyethylene terephthalate surfaces by a multistep approach was studied. In this approach, N-allylmethylamine was grafted onto a functionalized polyethylene terephthalate (PET) surface via air plasma treatment. Furcellaran, as a bioactive agent, was anchored on such substrates. Surface characteristics were measured by means of contact angle measurements, X-ray photoelectron spectroscopy (XPS) and scanning electron microscopy (SEM). Subsequently, samples were subjected to selected cell interaction assays, such as antibacterial activity, anticoagulant activity, fibroblasts and stem cell cytocompatibility, to investigate the Furcellaran potential in biomedical applications. Based on these results, furcellaran-coated PET films showed significantly improved embryonic stem cell (ESC) proliferation compared to the initial untreated material.

## 1. Introduction

Polysaccharides, in general, have increasingly attracted attention owing to their excellent antitumor, antioxidant, antibacterial, anti-inflammatory and anticoagulant activities [1]. Moreover, polysaccharides possess remarkable structural specifics for immobilization due to their strong binding to created brushes [2,3]. Coatings containing certain polysaccharides have improved cell adhesion and proliferation [4]. Furcellaran is a typical hybrid carrageenan extracted from the red algae *Furcellaria lumbricalis*. It is negatively charged and composed of units consisting of a fragment (1→3) β-_D_-galactopyranose with a sulfate group at C-4 and (1→4)-3,6-anhydro-α-_D_-galactopyranose. Structurally, furcellaran resembles the algal polysaccharide κ-carrageenan, with a major structural difference that only approximately 60% of the _D_-galactose units are 4-sulfated [5]. The number and position of ester sulfate groups and the content of 3,6-anhydrogalactose (3,6-AG) are the main differences that affect the properties of different carrageenans [6]. Anticoagulant activity and cell proliferation appear to decrease with the presence of a sulfate group in the G-6S position. The positional influence on the anticoagulant activity is ordered A-2, G-2 > G-4 > G-6. The order of the cytotoxicity of sulfate groups is supposed to be G-6 > G-4 > G-2 > A-2 [7]. Higher levels of ester sulfate results in lower solubility temperature and lower gel strength. One of the important characteristics of these polysaccharides is their ability to form gels in the presence of specific ions, which is associated with a molecular conformational change from the coil to helix [8] and is potentially able to interact with globular proteins, including bovine serum albumin [9] and β-lactoglobulin [10], and fibrous proteins, such as gelatine [11]. The interactions with oppositely charged macroions have gained much attention in biomedical applications. One of the most prominent examples are chitosan based polyelectrolyte complexes which have been revealed to be applicable in drug delivery systems [12,13]. The film-forming ability and the anionic nature of furcellaran also make it possible to establish a binary polyelectrolyte complex with chitosan to obtain binary biopolymer films [14,15]. The properties of biopolymer films can be enriched by the addition of active ingredients, such as plant extracts [16], essential oils [17] and nanomaterials [18].

Polymeric biomaterials that are intended for clinical use must have excellent mechanical properties, as well as adequate surface properties to ensure biocompatibility while interacting with living tissue. The current problem that can be encountered with biomaterials used for implants and other devices may be acute and chronic inflammatory responses, ultimately leading to fibrous capsule formation and thereby impairing normal tissue growth [19,20]. Furthermore, insufficient hemocompatibility due to surface-induced thrombosis or restenosis must be taken into account [21,22]. Further improvements of given materials rely on the precise control of cellular interactions in terms of cell adhesion and proliferation via modification of the biomaterial surface [23]. The surface modification of polymers consists of changes in surface energy, polarity, charge, topography, etc. [24,25].

Therefore, activation procedures have been carried out to make the polymer surface more hydrophilic and obtain specific features by introducing functional groups (carboxylic, hydroxyl, carbonyl, hydroperoxide, etc.) for further covalent bonding with selected agents [26]. The typical surface modification techniques used include chemical vapor deposition, wet chemical methods, UV light radiation, ozone-induced treatment and plasma exposure. Among the physical and chemical methods used to obtain required surfaces without affecting the bulk properties, plasma treatment is an appropriate and efficient technique [27,28,29]. In addition to the efficiency of immobilization, plasma treatment is able to enhance hydrophilicity and create a high density of functional groups on the surface without the use of toxic chemicals and heat processing. Thus, this method is suitable for many immobilization techniques with biologically active molecules and chemically unstable substrates (including polysaccharides) [30,31].

The plasma-treated polymer results in a negatively charged surface; therefore, subsequent immobilization of a potential anionic polysaccharide is challenging. The negative charge prevents the formation of covalent bonds between the molecule interfaces and polymer surface embedded radicals due to electrostatic repulsive forces. The binding affinity can be increased by reducing the negative charge of the polymer surface. One promising solution might be introducing more positively charged groups through mediators, such as N-allylmethylamine (MAAM). This liquid chemical is able to graft onto a plasma-treated polymer surface by a copolymerization process to create a polymer brush structure with a high density of positively charged amino groups via a radical “surface from” reaction in the gaseous phase [32,33]. Such amine-rich coatings have been studied to control cell behavior, demonstrating a positive effect on cell adhesion, proliferation, and differentiation for different cell types, such as osteoblast-like cells and fibroblasts [34]. Moreover, the advantage of using these functional groups is the possibility of exploiting them for the immobilization of bioactive molecules [35,36,37,38].

In the present study, poly(ethylene terephthalate) (PET) was used as a substrate due to its unique mechanical properties, biostability and its moderate inflammatory response. The Biostability of PET is mainly due to the presence of hydrophobic aromatic groups with high crystallinity which restricts hydrolytic breakdown. The low degradability of the PET offers enduring support over time along with ultimate performance during the patient’s lifetime. This can also surmount the problems of asynchronous degradation with new tissue regeneration and harmful end products of degraded polymers [39]. Hence, due to these favorable properties, this polyester has been found to be suitable in many biomedical applications, including vascular prostheses, artificial heart valve sewing cuffs, and sutures [40]. However, the hydrophobic nature of PET can be a disadvantage in the case of insufficient cellular interactions at the tissue/biomaterial interface. Furthermore, the biocompatibility of their surfaces is limited due to the lack of reactive functional groups and related long-term anti-thrombogenic demands for in vivo applications [40,41].

This research is the first study dedicated to the description of various furcellaran applications in biomedicine. Moreover, to the best of our knowledge, no study using furcellaran as a polysaccharide film layer for PET functionalization has been reported. The objective of this study is to represent a method for immobilizing furcellaran onto PET surfaces. For this purpose, RF plasma discharge is applied onto the PET surface to create oxidative functional groups for further covalent binding with selected agents. First, MAAM is grafted onto plasma-treated PET to create a high-density polymer brush for improved adhesion interaction properties. Consequently, furcellaran immobilization on such a treated surface is performed. As-prepared substrates are subjected to various selected cell interactions. Namely, the antibacterial activity, anticoagulant activity, fibroblasts and stem cell cytocompatibility of the samples were investigated.

## 2. Materials and Methods

### 2.1. Materials and Preparation of PET Films

Polyethylene terephthalate (PET) in rectangle form sheets (55 × 30 mm) was rinsed with distilled water and subsequently dried at room temperature. Monomer N-allylmethylamine (MAAM) was purchased from Sigma–Aldrich (St. Louis, MO, USA). κ-carrageenan (κ-CA) and furcellaran (FUR) of different water gel strengths (Estgel 1000, Estgel 8500) were obtained from Est-Agar AS (Karla, Estonia). The solution was prepared by dissolving 0.1% (*w*/*v*) furcellaran and κ-carrageenan in distilled water. Diiodomethane (99.0%, reagentplus) and formamide (99.5%, molecular biology grade) were supplied by Sigma–Aldrich (St. Louis, MO, USA).

### 2.2. Plasma Surface Modification and MAAM Grafting

Both sides of the PET films were treated by low-pressure plasma equipment (Diener Electronic, Nagold, Germany) for 60 s at a radio frequency of 13.56 MHz. The discharge matching power was set to 50 W, and the applied air feed rate was 20 sccm (standard cubic centimeter per minute). The pressure inside the chamber was approximately 60 Pa. Some of the plasma-treated PET samples were subsequently immediately exposed to saturated MAAM vapors to achieve radical graft monomer polymerization towards polymer brushes, which are suitable for biological agent immobilization (including polysaccharides). For comparison, the characterization and ensuing treatments of PET samples were also performed without previous MAAM grafting (Figure 1).

### 2.3. Furcerallan and κ-Carrageenan Immobilization

MAAM grafted and nongrafted PET samples were immersed into a 0.1% (*w*/*v*) solution of κ-CA and FUR for 24 h at room temperature for their immobilization onto a previously prepared polymer brush. After deposition, the samples were removed from the solution and washed in distilled water to remove residual unbound polysaccharide content. Finally, the samples with immobilized polysaccharide layers were dried overnight at room temperature.

### 2.4. Contact Angle Measurement and Surface Energy Evaluation

The surface wettability was evaluated by the water sessile drop contact angle method via a SEE system (by Advex Instruments, Brno, Czech Republic). Lewis acid-base parameters of the surface free energy of the samples were evaluated from contact angle values using appropriate testing liquids, i.e., water, diiodomethane, formamide [42]. Five separate readings for each liquid were averaged to obtain representative contact angle values, which were used for surface free energy evaluation for each surface. 

Droplets of testing liquid with a volume of 5 µL were placed onto each sample and captured by a CCD camera system. This droplet volume was set for several reasons. The most important factor was the fact that larger drops are deformed by gravity. Smaller drops give results with higher measurement errors.

### 2.5. X-ray Photoelectron Spectroscopy

The chemical composition was analyzed with X-ray photoelectron spectroscopy (XPS) using TFA (Physical Electronics, Chanhassen, MN, USA) with MultiPak software to determine the concentration of elements. The samples were irradiated with X-rays with a 400 µm spot size generated with monochromatic Al Kα1,2 radiation at 1486.6 eV. The emitted photoelectrons were detected with a hemispherical analyzer placed at an angle of 45° relative to the normal plane of the sample surfaces.

### 2.6. SEM Evaluation

The surface morphology of each sample was monitored by a NANOSEM 450 (FEI, Thermo Fisher Scientific, Hillsboro, OR, USA) scanning electron microscope operated at 5 kV with an Everhart–Thomley detector (ETD). The images were obtained at a magnification of 20,000×. The samples were coated with Au/Pt prior to observation.

### 2.7. Evaluation of Antibacterial Activity

The antibacterial activity was determined according to a modified version of the ISO 22196 standard for investigation of the antibacterial effect on modified plastic materials. First, all samples were sterilized by UV radiation and then placed in sterile Petri dishes. Standardized bacterial suspensions of *Escherichia coli* (CCM 4517) and *Staphylococcus aureus* (CCM 2022) were prepared in 1/500 nutrient broth and diluted to obtain a certain bacterial concentration between 2.5 × 10^5^ cells/ml and 10 × 10^5^ cells/ml. Thus, the prepared solutions were used as the test inoculum, which was dispensed onto each sample (25 × 25 mm) in a volume of 0.4 mL. Subsequently, the samples were covered with ethanol-disinfected polypropylene foil (PP) (20 × 20 mm^2^) and incubated at 35 °C and 95% relative humidity for 24 h. After the incubation period, each sample with PP films was washed with 10 mL of SCDLP broth (HiMedia Laboratories, Mumbai, India) to determine the recovery rate of the bacteria. The recovered bacterial suspensions were subjected to 10-fold serial dilutions. The viable bacteria count was evaluated by the pour plate culture method after 24 h of incubation.

### 2.8. Evaluation of Anticoagulant Activity

Blood was obtained by venous puncture from a healthy donor in accordance with the Helsinki Declaration and placed into blood collection tubes (VACUETTE, Greiner Bio-One, Kremsmünster, Austria) covered by prepared PET samples. The obtained human blood plasma was treated with 3.2% citric acid (109 mmol/L) and then centrifuged at room temperature for 15 min at 3000 min. The anticoagulant activity was determined by means of prothrombin time (PT), thrombin time (TT) and activated partial thromboplastin time (aPTT) using a SYSMEX CA-1500 (Siemens, Munich, Germany) instrument. Each of the samples was examined three times.

### 2.9. In Vitro Fibroblast Cytocompatibility

The cytotoxicity was tested using a mouse embryonic fibroblast cell line (NIH/3T3, ATCC® CRL-1658TM, Manassas, VA, USA) according to the EN ISO 10993-5 standard with some modifications. The tested samples with dimensions of 10 × 10 mm were exposed to UV radiation (wavelength of 253.7 nm; 30 min) for sterilization. ATCC-formulated Dulbecco’s modified Eagle’s medium (BioSera, Nuaille, France) containing 10% calf serum (BioSera, Nuaille, France) and 100 U mL^−1^ penicillin/streptomycin (BioSera, Nuaille, France) was used as the culture medium. The cells were seeded onto samples at a concentration of 2 × 10^4^ cells per mm^2^ and incubated at 37 ± 1 °C for 72 h. After 72 h of cell culture, the cell viability was evaluated using the 3-(4,5-dimethylthiazol-2-yl)-2,5-diphenyltetrazolium bromide (MTT) assay (Duchefa Biochemie, Amsterdam, The Netherlands). First, the culture medium was removed, and 100 µL of growth medium containing MTT dye solution (5 mg/mL in PBS) was added to the cultures. Then, the cells were incubated at 37 °C in a humidified atmosphere for 4 h. Following the removal of the growth medium, DMSO (Sigma–Aldrich, St. Louis, MO, USA) was added to dissolve the formed formazan crystals on the sample surface. The absorbance was measured using an Infinite M200 Pro NanoQuant absorbance reader (Tecan, Männedorf, Switzerland) at wavelengths of 570 nm (test) and 690 nm (reference). The reported values are the means of three replicates and are expressed as percentages of the control value.

### 2.10. In Vitro Stem Cell Cytocompatibility

The embryonic stem cell line ES R1 was propagated in an undifferentiated state by culturing on two series of prepared samples. One series was coated with 0.1% gelatine, and the other was not. Dulbecco’s Modified Eagle’s Medium with high glucose, pyruvate, 16.5% calf serum (all from Gibco, Thermo Fisher Scientific, Inc., Waltham, MA, USA), 1% penicillin/streptomycin (GE Healthcare HyClone, HyClone Ltd., Cramlington, UK), 100 mM nonessential amino acids (Gibco, Thermo Fisher Scientific, Inc., Waltham, MA, USA), 0.05 mM b-mercaptoethanol (Sigma–Aldrich, St. Louis, MO, USA), and leukemia inhibitory factor (Chemicon International, Temecula, CA, USA) at a concentration of 5 ng.mL-1 was used as the culture medium. First, both sides of the prepared samples were sterilized by UV light for 30 min. Then, the cell suspension was seeded on the tested samples at a concentration of 20,000 cells per mL and incubated at 37 °C in 5% CO_2_ in humidified air. After 120 h (5 days) of proliferation, the cells were fixed and stained to visualize the DNA with Hoechst 33258 (Molecular Probes, Carlsbad, CA, USA) and observed by fluorescence microscopy using an Olympus IX 81 inverted phase-contrast microscope (Olympus, Hamburg, Germany). An MTT proliferation test was performed. All tests were performed in four separate sets. Statistical significance was determined by ANOVA with post hoc Tukey’s multiple comparison test, * *p* < 0.05, ** *p* < 0.01, *** *p* < 0.001

## 3. Results and Discussion

### 3.1. Surface Wettability Investigations

Wettability changes resulting from surface modifications were determined from contact angle data for various testing liquids and are shown in Table 1. The hydrophobic nature and lack of functional groups of untreated PET are reflected in the high values of the contact angle. After air plasma treatment, the contact angle values sharply decreased due to the presence of oxidative functional groups introduced onto the PET surface and tailoring of the surface morphology, referring to the increased hydrophilicity and surface energy suitable for further immobilization. The hydrophobicity of immobilized FUR and κ-CA samples grafted onto polymer brushes increased but remained less hydrophobic than untreated PET. In comparison to the films without MAAM, no noticeable change occurred. In addition, modification effects on the surface were observed with respect to the Lewis acid-base properties of the samples. For the γ_s_ of untreated PET foil, low values were calculated due to its low surface wettability. The value of γ_s_ after air plasma exposure seemed to be distinctly higher, indicating that the surface had a relatively significant polarity. Variations between the γ_s_ values of plasma-treated PET and immobilized PET were not significant. The results indicate successful immobilization of FUR and κ-CAR onto the PET surface, as subsequently demonstrated in detail by XPS.

### 3.2. Surface Chemistry Investigations

The prepared samples with the different surface treatments were investigated by the XPS method, and their elemental compositions are shown in Table 2. As expected, the surface of reference PET revealed a maximum content of carbon of 74.6% and an oxygen level of 25.4%. The chemical composition induced by plasma treatment showed a considerable increase in the level of oxygen due to the incorporation of oxidized functionalities. Nitrogen content of 1.5% might be formed during the plasma treatment process. Nevertheless, the highest contents of nitrogen were observed for MAAM-grafted samples as a result of the amine groups present in the reagent, but a proportional decrease in the oxygen level was noted. In contrast, the specimens that were not treated with MAAM had slightly higher oxygen levels. The most important result was the sulfur content presence, which was detected for the samples MAAM_KAPA, DC_1000, DC_8500 and DC_KAPA. The sulfur and oxygen contents indicate the successful immobilization of polysaccharides despite the sulfur quantities being relatively low.

### 3.3. SEM Imaging

The surface morphology was examined by means of SEM performed with an Everhart–Thomley detector without a conductive coating of the samples and represented at a 10 μm resolution. The results are shown in Figure 2a–i, where the reference PET exhibits a homogenous, relatively smooth surface morphology. Among the polysaccharide-immobilized surfaces, the furcellaran-coated PET (Figure 2d,e,g,h) showed a more homogenously distributed layer, which allowed uniform adhesion of fibroblasts and their proliferation on the surface [43]. In contrast, the PET surface coated with κ-carrageenan (Figure 2i) had a rather heterogeneous nature, which may alter the behavior of cells adhering to the substrates. Similarly, an inhomogeneous and rough morphology was observed on the MAAM-grafted PET surface, where such characteristics are desired for further immobilization [44].

### 3.4. Antibacterial Analysis

The antibacterial activity levels of PET films against *Staphylococcus aureus* and *Escherichia coli* strains were evaluated by the number of viable cells calculated in the agar plates after 24 h of incubation. As shown in Table 3, a very low antibacterial effect was displayed by the immobilized samples compared to the PET reference. There are few reports about the antimicrobial effects of carrageenans. Yamashita et al. evaluated the antimicrobial actions of three types of carrageenans (ι, λ, κ) and showed their significant inhibitory effect on almost all bacterial strains studied. The study also indicated that the removal of sulfate residues eliminates the bacteriostatic effect of ι-carrageenan, suggesting that the sulfate residues in carrageenans play an essential role in this effect. Furthermore, the addition of carrageenan significantly lowered the growth rate of bacteria in a concentration-dependent manner, although the effect was bacteriostatic rather than bactericidal. However, the inhibitory activity is unlikely to have been dependent on the sulfate content alone but on the inhibitory mechanism itself, which needs to be studied further. Another study [45] showed that oxidized κ-carrageenan could suppress the growth of both gram-positive and gram-negative bacteria. However, this was not the case for natural κ-carrageenan, which does not exhibit any antibacterial activity. PVA-κ-carrageenan films crosslinked with glutaraldehyde have no effect on the antibacterial performance unless they are loaded with antibacterial drugs [46]. Based on the mentioned studies, the lower sulfate content of furcellaran and the unmodified nature of immobilized polysaccharides overall may contribute only to the reduced growth-inhibitory effect. In addition, the amounts of immobilized furcellaran and κ-carrageenan on the PET surface associated with the minimum inhibitory concentration (MIC) might also be the reason for the insufficient antibacterial activity.

Along with the presence of the sulfate group, the antibacterial activity of polysaccharides can be enhanced via chemical modification by incorporating other functional groups, such as carboxylic and carboxymethyl groups, aldehyde, amine, alkylamine, and quaternary ammonium groups. The hydroxyl groups can be also grafted with organic moieties offering extra functional groups to improve bioactivities according to the given application [47,48]. Based on these results, furcellaran is not considered as promising an antibacterial component as some natural polysaccharides, for instance, chitosan [49], chondroitin sulfate [27] or fucoidan [50].

### 3.5. Anticoagulant Activity Assessment

The anticoagulant activity of PET samples was evaluated by thrombin time (TT), prothrombin time (PT) and activated partial thromboplastin time (aPTT) coagulation assays, and the obtained results are summarized in Table 4.

In general, the blood coagulation system is divided into two initiating pathways: the tissue factor (extrinsic) pathway and the contact factor (intrinsic) pathway. These pathways meet in a final common pathway where factor Xa converts prothrombin to thrombin, which then cleaves fibrinogen to fibrin monomers.

PT is a functional measure of the extrinsic and common pathways, and the physiological value is in the range of 11–13.5 s. As seen, all samples are within the range, meaning that the extrinsic pathway of coagulation was not inhibited by any of the tested samples. These data are consistent with a study on the anticoagulant activity of major types of carrageenans [6].

aPPT is a functional measure of the intrinsic pathway as well as the common pathway, and the reference range is between 22 and 33 s. As the results show, the clotting times of samples PET_DC_8500 and PET_DC_KAPA were slightly prolonged, and therefore, interfered with the intrinsic coagulation process.

TT is a measure of functional fibrinogen in plasma, and for healthy donors, the clotting time range is between 12.5 and 19 s. PET_DC_KAPA displayed the highest value of clotting time and thereby mildly exceeded the threshold for anticoagulant activity. As follows, plasma-treated PET exhibited a prolonged time over the threshold. The hydrophilic surface achieved by plasma treatment may be associated with lower protein adsorption and the activation of blood components. Another factor influencing protein adsorption on hydrophilic surfaces is the type of charge. Negatively charged surfaces typically lead to a lower extent of protein adsorption than positively charged surfaces because many proteins have a net negative charge at physiological pH values [51]. These reports correspond with the fact that the presence of sulfate groups in furcellaran and κ-carrageenan might be attributed to the change in the negative charge density and influence its interactions with the coagulation proteins.

Thus, higher values were observed in plasma-treated PET and polysaccharide-immobilized samples in combination with plasma treatment, which correlated with the content of sulfates and hydrophilic surfaces [47,48,49,50]. However, samples treated with MAAM had a negative impact on anticoagulant activity. It is worth noting that none of the samples extended beyond anticoagulation action. In many cases, the prolongation of clotting time is dose-dependent [51,52]. With respect to the abovementioned, a slight anticoagulant effect was observed in the case of sample PET_DC_KAPA. However, it must be taken into account that the effect is not as significant as in the case of other sulfated polysaccharides [28].

### 3.6. In Vitro Fibroblast Cytocompatibility

The cytotoxicity of individual samples was evaluated using a mouse embryonic fibroblast cell line (NIH/3T3) according to the ISO 10993-5 standard. Cytotoxicity tests are the first option when the biocompatibility of materials, including polymers, is assessed. Measurements were performed in triplicate with untreated PET as a reference. Error bars represent standard deviations. Figure 3 shows that all values obtained from the relative cytotoxicity test were above 80%. Overall, the cell viability of samples functionalized with MAAM and polysaccharides (FUR, κ-CA) increased in comparison with samples treated only with plasma, and all materials can be considered nontoxic. This was also confirmed in Figure 4a–i, where cells were growing homogeneously, and all samples were assigned as cytocompatible. The biological response to any material is affected by its various surface properties. Hydrophobicity, roughness, homogeneity or functional groups belong to important factors affecting cell adhesion and proliferation. Many studies indicated that cells tended to attach to hydrophilic surfaces [44]. On the other side, it has been reported that cells adhered and proliferated at the highest rate when cultured on a hydrophobic surface or a contact angle of around 70 degrees [53]. This may be the explanation for why untreated PET showed the best proliferation results. As expected, the cell proliferation effect of furcellaran coated samples is consistent with previous studies of other seaweed polysaccharides [54,55,56].

### 3.7. Evaluation of Cell Viability in the ES R1 Cell Line

Examples of versatile, non-animal derived and inexpensive materials that are able to support the proliferation of ESCs are limited. Many of the new biomaterials used to develop stem cell microenvironments use biopolymers adsorbed or covalently immobilized to the surface to improve the biocompatibility of synthetic polymers [57]. Furcellaran coated PET samples were suggested as a potential candidate to promote the proliferation of ESC. Embryonic stem cells proliferated on uncoated and gelatine-coated samples for 5 days. As shown in Figure 5A, there was no significant difference in cell viability between the reference (cells cultivated on TPP plastic) and PET. Figure 5B shows cell viability on TPP plastic coated with gelatine (reference) and PET coated with gelatine. The results did not indicate a significant difference in cell viability. Therefore, PET was deemed to be a good material for cell cultivation and was considered a reference in subsequent testing (Figure 5C,D). The polymerization of N-allylmethylamine (MAAM) onto plasma-treated PET films was performed to introduce amine (-NH_2_) functional groups and reveal the optimum polymer brush for the immobilization of the tested polysaccharides. Amine groups are known to promote cell adhesion because of their positive charges, which can attract negatively charged biomolecules, such as proteins under physiological conditions. It was expected that MAAM-treated samples would demonstrate increased cell proliferation due to a higher quantity of polysaccharide agents immobilized. As shown in Figure 5C, there was a significant reduction in cell viability on sample MAAM_8500. One possible explanation for this observation is the correlation between the gel strength of FUR 8500 and insufficient interaction with the amine groups of MAAM. The gel strength of FUR 8500 was lower than that of FUR 1000, which also corresponded to the lower content of 3,6-anhydro-D-galactose residues. The amount of these units could adversely affect the immobilization of FUR 8500. In general, the existence of MAAM polymer brushes was revealed; however, intramolecular interactions between grafted polymer brushes and polysaccharides might not be sufficient for immobilization, and therefore, negatively influence subsequent interactions with ESC. However, the most important result is that for samples DC_8500 (Figure 6B) and DC_KAPA, ESC proliferation was significantly increased compared to that of the reference PET (Figure 5A). This might be a result of sufficient polysaccharide immobilization. Thus, more polysaccharide on the surface is able to enhance the proliferation of ESCs. All of the samples coated with gelatine (Figure 6D) did not show a cytotoxic effect, except sample MAAM_8500_G. 

## 4. Conclusions

In this work, furcellaran deposition onto a PET surface via RF air plasma discharge activation followed by grafting of a MAAM monomer was studied. The successful immobilization of polysaccharides onto grafted and nongrafted MAAM monomer brushes was confirmed by the contact angle and XPS results, where the surface wettability of the resultant films increased compared to the PET reference, and the sulfur content characteristic of the tested polysaccharides was detected. SEM images of coated PET samples demonstrated a rather microtextured homogenously distributed layer, which had positive effects on the uniform adhesion and proliferation of fibroblasts and thus exhibited nontoxicity for all functionalized samples. All samples had minor effects on the antibacterial activity and blood coagulation, as indicated by the clotting times of APPT, PT and TT. 

However, MAAM nongrafted samples of immobilized FUR significantly increased ESC proliferation, in contrast to the MAAM grafted ones. The low ESC proliferation rate for MAAM grafted samples might be due to the insufficient interaction of polysaccharides with amine groups. The obtained data for furcellaran were compared with the results for κ-carrageenan. These preliminary results show that furcellaran itself has significant biomedical application prospects and paves the way for future investigations.

## Figures and Tables

**Figure 1 ijms-23-07439-f001:**
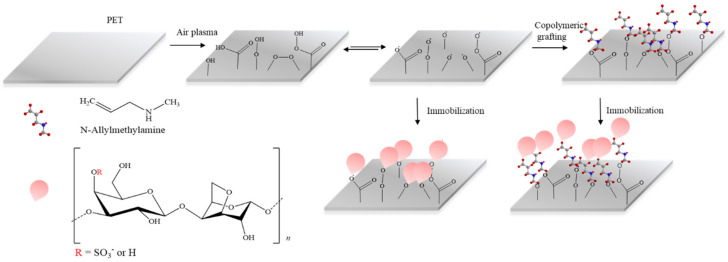
Schematic representation of plasma postirradiation grafting of N-allylmethylamine onto a PET surface followed by immobilization of FUR or κ-CA polysaccharide.

**Figure 2 ijms-23-07439-f002:**
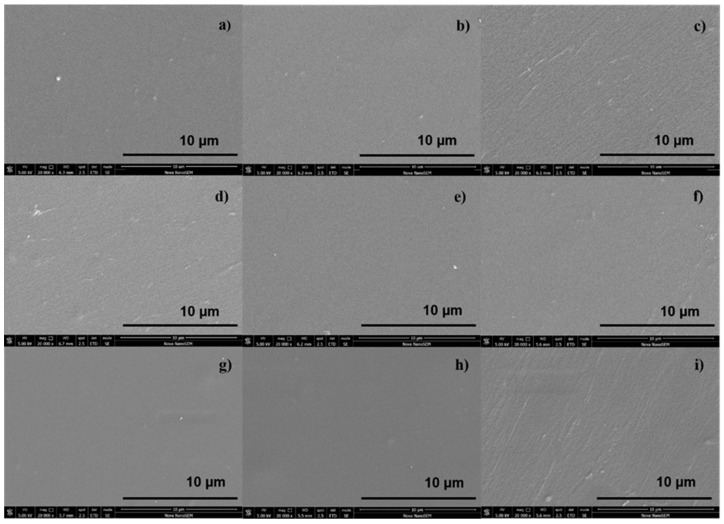
SEM images of (**a**) untreated PET, (**b**) PET_DC, (**c**) DC_MAAM, (**d**) MAAM_1000, (**e**) MAAM_8500, (**f**) MAAM_KAPA, (**g**) DC_1000, (**h**) DC_8500 and (**i**) DC_KAPA.

**Figure 3 ijms-23-07439-f003:**
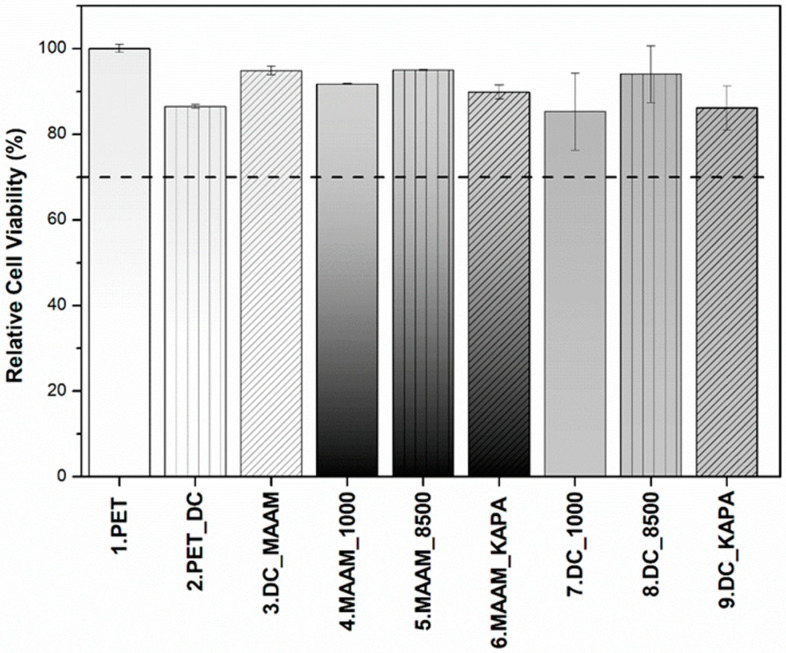
Relative cell viability of cells incubated with PET films coated with furcellaran of different water gel strengths (1000, 8500) and κ-carrageenan tested on a mouse embryonic fibroblast cell line (NIH/3T3).

**Figure 4 ijms-23-07439-f004:**
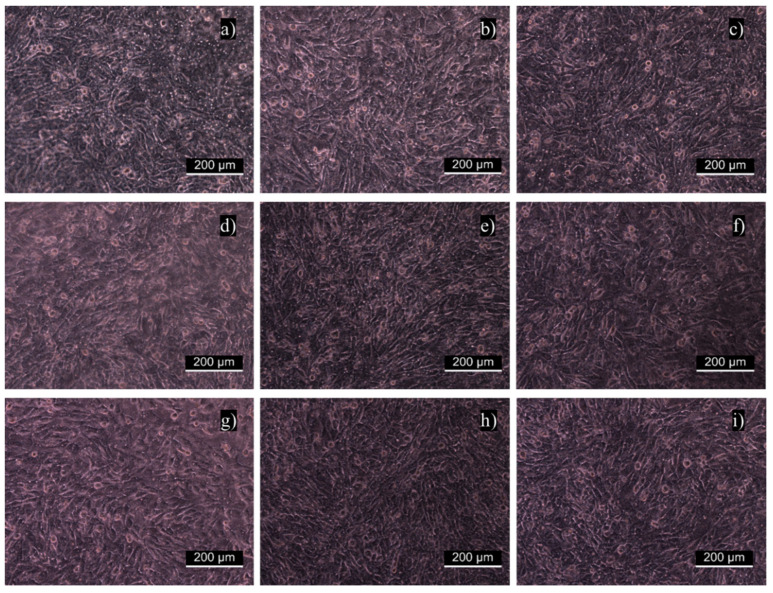
Fibroblast proliferation on examined samples: (**a**) untreated PET, (**b**) PET_DC, (**c**) DC_MAAM, (**d**) MAAM_1000, (**e**) MAAM_8500, (**f**) MAAM_KAPA, (**g**) DC_1000, (**h**) DC_8500 and (**i**) DC_KAPA.

**Figure 5 ijms-23-07439-f005:**
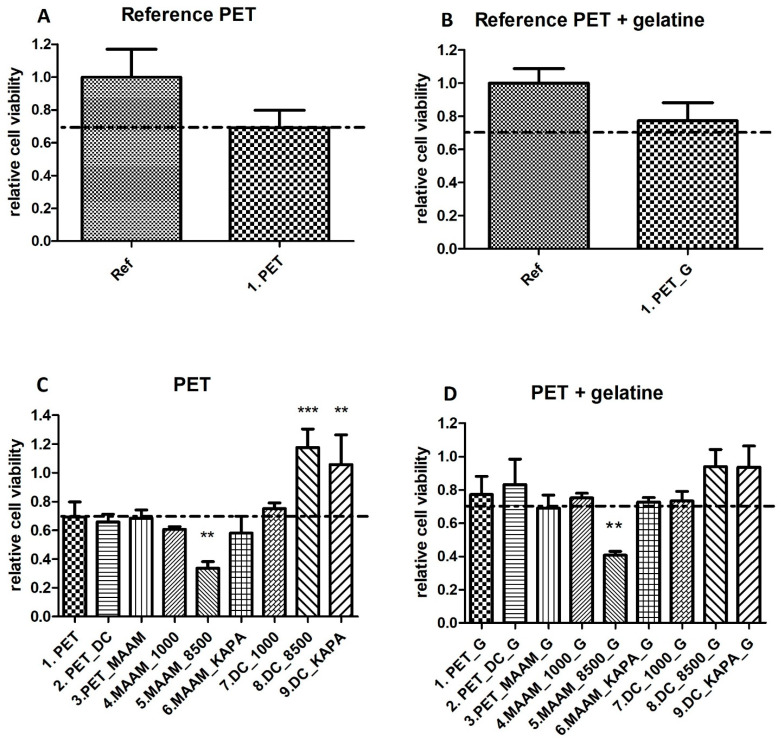
Cell viability of individual samples. The dashed line highlights the limit of viability according to EN ISO 10993-5: viability < 0.7 corresponds to a cytotoxic effect. (**A**) Comparison of reference (cells on TPP plastic) and PET; (**B**) Comparison of reference (cells on TPP plastic coated with gelatine) and PET coated with gelatine; (**C**) Comparison of PET as a reference and samples; (**D**) Comparison of PET coated with gelatine as a reference and samples also coated with gelatine. Data are reported as the means and standard deviations from a minimum of four independent experiments. To determine significant differences between samples ANOVA with post hoc Tukey’s Multiple Comparison test was used; ** *p* < 0.01, *** *p* < 0.001.

**Figure 6 ijms-23-07439-f006:**
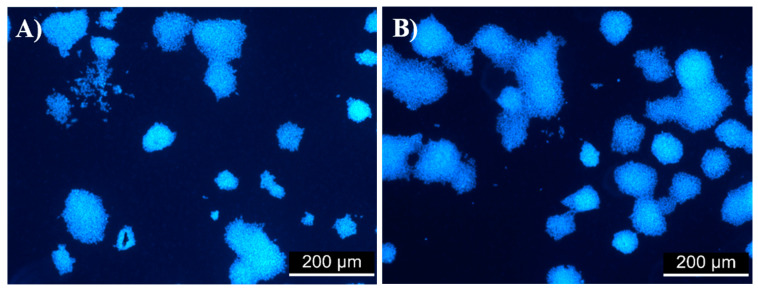
Mouse embryonic cells (Line R1) on (**A**) PET uncoated with 0.1% gelatine and (**B**) DC 8500 uncoated with 0.1% gelatine after 120 h of cultivation.

**Table 1 ijms-23-07439-t001:** Contact angles (θ) (w: deionized water; d: diiodomethane; f: formamide) and surface free energy parameters of probe liquids used in the acid-base method (γg total surface fee energy, apolar γsLW, polar γsAB, Lewis acid γs+ and base γs- components). PET is the virgin polyethylene terephthalate sample, PET_DC is the PET sample treated in discharge and PET_MAAM is the PET_DC sample after grafting with N-allylmethylamine. MAAM_1000, MAAM_8500 and MAAM_KAPA are samples in which selected polysaccharides were immobilized on the PET_MAAM sample. DC_1000, DC_8500 and DC_KAPA indicate samples in which selected polysaccharides were immobilized on the PET_DC sample.

	Contact Angles (°) for Liquids			Surface Free Energy Parameters (mJ/m^2^)				
SAMPLE	θ_w_	θ_d_	θ_f_	γ_s_	γ_s_^LW^	γ_s_^AB^	γ_s_^+^	γ_s_^−^
PET	63.27 ± 3.08	26.63 ± 0.54	53.72 ± 0.95	50.17	45.55	4.62	0.24	22.10
PET_DC	30.25 ± 3.40	25.57 ± 0.61	8.97 ± 1.15	57.64	45.95	11.72	0.88	38.80
PET_MAAM	33.71 ± 1.83	30.76 ± 0.38	12.22 ± 2.62	57.08	43.90	13.18	1.22	35.70
MAAM_1000	40.94 ± 1.74	27.78 ± 2.91	17.8 ± 2.81	56.28	45.11	11.17	1.07	29.27
MAAM_8500	47.33 ± 1.95	32.02 ± 2.36	20.18 ± 2.05	55.32	43.36	11.96	1.59	22.52
MAAM_KAPA	49.19 ± 2.0	28.16 ± 1.15	18.73 ± 2.28	56.11	44.98	11.13	1.55	20.02
DC_1000	36.75 ± 1.35	18.06 ± 1.55	12.08 ± 2.03	57.98	48.33	9.65	0.71	32.62
DC_8500	47.08 ± 2.25	21.67 ± 0.80	10.99 ± 1.46	58.35	47.27	11.07	1.49	20.61
DC_KAPA	46.71 ± 0.99	26.82 ± 1.94	11.28 ± 1.69	57.72	45.48	12.24	1.79	20.97

**Table 2 ijms-23-07439-t002:** Surface elemental compositions of the samples (%) from XPS measurements.

SAMPLE	C	N	O	S
PET	74.6		25.4	
PET_DC	69.3	1.5	29.3	
PET_MAAM	69.8	2.1	28.1	
MAAM_8500	73.7	1.0	25.3	
MAAM_1000	74.6	1.0	24.5	
MAAM_KAPA	74.2	0.9	24.8	0.2
DC_1000	70.0		29.7	0.3
DC_8500	73.9	0.6	25.5	0.1
DC_KAPA	72.7	0.7	26.5	0.1

**Table 3 ijms-23-07439-t003:** Viable bacteria numbers on PET surfaces; antibacterial effect (R) values are expressed in CFU.

SAMPLE	*Staphylococcus aureus* CCM 2020		*Escherichia coli* CCM 4517	
	N (cfu/cm^2^)	R	N (cfu/cm^2^)	R
PET	1.1 × 10^4^	0	9.8 × 10^5^	0
PET_DC	1.1 × 10^4^	−0.02	8.9 × 10^5^	0.05
PET_MAAM	1.1 × 10^4^	−0.02	8.3 × 10^5^	0.07
MAAM_1000	1.2 × 10^4^	−0.04	5.8 × 10^5^	0.22
MAAM_8500	1.2 × 10^4^	−0.05	8.6 × 10^5^	0.05
MAAM_KAPA	4.6 × 10^3^	0.38	9.5 × 10^5^	0.01
DC_1000	8.1 × 10^3^	0.14	8.2 × 10^5^	0.07
DC_8500	2.4 × 10^4^	−0.34	7.1 × 10^5^	0.14
DC_KAPA	8.6 × 10^3^	0.11	5.8 × 10^5^	0.12

**Table 4 ijms-23-07439-t004:** Anticoagulation activity results; PT: prothrombin time; aPTT: activated partial thromboplastin time; TT: thrombin time.

SAMPLE	PT (s)	aPTT (s)	TT (s)
PET	12.0	29.1	18.0
PET_DC	12.6	32.8	20.1
PET_MAAM	12.6	30.5	18.5
MAAM_1000	12.1	28.8	18.4
MAAM_8500	12.4	30.6	18.6
MAAM_KAPA	12.3	29.7	18.9
DC_1000	13.0	32.5	18.7
DC_8500	13.3	33.4	19.7
DC_KAPA	12.3	33.9	20.5

## Data Availability

The data presented in this study are available upon reasonable request from the corresponding author.

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
