# Peer review of "Furcellaran Surface Deposition and Its Potential in Biomedical Applications"

_ijms, 2022, doi:10.3390/ijms23137439_

Round 1

Reviewer 1 Report

The authors reported the antibacterial activity for plasma postirradiation grafting of N-allylmethylamine onto a  PET surface followed by immobilization of FUR or κ-CA polysaccharide. The submission requires revision before acceptance, considering the following points:-

1.      The materials characterization should be improved by measuring XRD, FT-IR, XPS, …etc.

2.      Results and discussion should be rearranged to be understandable. For example, Figure 5 should be Figure 1. The data representation should be rearranged logically.

3.      The challenges such as low PET degradability should be considered and highlighted in the study.

4.      Figure 5 should be confirmed using experimental data to support the author’s claims.

5.      A comparison with previously published results should be discussed.

6.      The References for the antibacterial activity for other materials should be updated, including these References; https://doi.org/10.3390/ijms23105405; doi: 10.3389/fceng.2021.790314

7.      The language should be improved, and typos should be corrected.

Minors

8.      Correct typos such as ‘2 × 104 cells per mm2 and incubated’; ‘100 U mL−1’; ‘10 x10 mm’; 

Author Response

REVIEWER 1:Comments and Suggestions for Authors

The authors reported the antibacterial activity for plasma postirradiation grafting of N-allylmethylamine onto a  PET surface followed by immobilization of FUR or κ-CA polysaccharide. The submission requires revision before acceptance, considering the following points:-

Q1. The materials characterization should be improved by measuring XRD, FT-IR, XPS, …etc.

A1: Many thanks. The results obtained by XPS are mentioned in the text (table 2, lines 265-280)

Q2. Results and discussion should be rearranged to be understandable. For example, Figure 5 should be Figure 1. The data representation should be rearranged logically.

A2: Thank you for your comment on the confusing layout of the Results and Discussion section that resulted from the conversion to the template.The data has been represented coherently.

Q 3. The challenges such as low PET degradability should be considered and highlighted in the study.

A3: Thank you for highlighting the information related to the PET degradability. This information has already been added, as suggested.

Q 4. Figure 5 should be confirmed using experimental data to support the author’s claims.

A4: The Figure 5 (now Fig. 1)  is a consequence of obtained XPS results as well as knowledge obtained during previous studies of surface biofunctionalization. The fig. 1  is in agreement with XPS results.

Q5. A comparison with previously published results should be discussed.

A5: Thank you. As suggested, we implemented comparison with other polysacharides.

Q6. The References for the antibacterial activity for other materials should be updated, including these References; https://doi.org/10.3390/ijms23105405; doi: 10.3389/fceng.2021.790314

A6: Many thanks for relevant references. We implemented both into the manuscript.

Q7. The language should be improved, and typos should be corrected.

A7: We went thoroughly across the manuscript again and we corrected some typos.

Q8. Correct typos such as ‘2 × 104 cells per mm2 and incubated’; ‘100 U mL−1’; ‘10 x10 mm’; 

A8:  Thank you for the notification The typos mentioned above have been fixed.

All in all, thank you so much for your suggestions which made the manuscript of much higher quality.

Reviewer 2 Report

Dear authors,

I find your work very interesting and with a high impact on the field of biomedicine. However, I consider that this paper may be improved before it is published as follows:

1. In the Introduction (line 52) was mentioned that "These interactions with oppositely charged macroions have gained much attention in biomedical applications". Few examples may be added. 

2. Figure 1 does not show the SEM images for all tested surfaces.

3. In line 98 it is stated that "the hydrophobic nature of PET can be a disadvantage in the case of insufficient cellular interactions at the tissue/biomaterial interface", but how do you explain the fact that the highest cell viability was obtained on this untreated surface? (Figure 2). 

4. Given that microscopy images are more relevant for cell adhesion and proliferation on various surfaces, images obtained for the NIH/3T3 murine fibroblast line should also be included in the article.

5. Why is Figure 3 showing only 2 of the tested surfaces? 

6. In Materials and Methods section (line 392) it is mentioned that "the cells were observed by fluorescence microscopy 392 using an Olympus IX inverted phase contrast microscope". What staining method/fluorochrome was used? This information is missing.

7. The conclusions can be extended. It is not very clear which surfaces have the greatest biomedical potential and why.

8. In line 258 I think it is not Figure 1C. More care in writing (there are small mistakes in the entire manuscript).

Author Response

REVIEWER 2:Comments and Suggestions for Authors

Dear authors,

I find your work very interesting and with a high impact on the field of biomedicine. However, I consider that this paper may be improved before it is published as follows:

Q1. In the Introduction (line 52) was mentioned that "These interactions with oppositely charged macroions have gained much attention in biomedical applications". Few examples may be added. 

A1: Thank you for proposing to add an example regarding oppositely charged macroions and their use in biomedicine. We have added one well-known example with use in drug delivery.

Q2. Figure 1 does not show the SEM images for all tested surfaces.

A2: For Fig. 1 (now Fig.2), there are present images for all tested surfaces, as recommended. Many thanks for your remark.

Q3. In line 98 it is stated that "the hydrophobic nature of PET can be a disadvantage in the case of insufficient cellular interactions at the tissue/biomaterial interface", but how do you explain the fact that the highest cell viability was obtained on this untreated surface? (Figure 2). 

A3: This information has been reported in (10.1016/j.msec.2016.08.065).Many studies indicated that cells tended to attach onto hydrophilic surfaces. On the other side, it has been reported that cells adhered and proliferated at the highest rate when cultured on a hydrophobic surface. This information is already implemented into the manuscript. In fact, there is not of a paramount importance which sample is of the highest cell viability. The most important result is that samples are above the accepted level of 80 % cytotoxicity what means that the sample is of enough cappability to be used in biomedicine.

Q4. Given that microscopy images are more relevant for cell adhesion and proliferation on various surfaces, images obtained for the NIH/3T3 murine fibroblast line should also be included in the article.

A4: As suggested, we present now all images of tested samples for the NIH/3T3 murine fibroblast line. Thank you for suggestion.

Q5. Why is Figure 3 showing only 2 of the tested surfaces? 

A5: Thank you. We presented the most evident results on representative images. Nevertheless, as recommended, we present now all tested samples. However, the most important are data presented in Figure 5 which are results obtained by fluorescence microscopy and statistical significance was determined by ANOVA with post hoc Tukey’s multiple comparison test.

Q6. In Materials and Methods section (line 392) it is mentioned that "the cells were observed by fluorescence microscopy 392 using an Olympus IX inverted phase contrast microscope". What staining method/fluorochrome was used? This information is missing.

A6: Thank you for notifying the missing information, which has  been  already added.

Q7. The conclusions can be extended. It is not very clear which surfaces have the greatest biomedical potential and why.

A7: We have tried to broaden the conclusion for a better understanding.

Q8. In line 258 I think it is not Figure 1C. More care in writing (there are small mistakes in the entire manuscript).

A8: Many thanks for reporting some inconsistencies in the text.The mistakes have been corrected.

All in all, thank you so much for your suggestions which made the manuscript of much higher quality.

Round 2

Reviewer 1 Report

The authors addressed most of the comments and the revised version can be accepted.

Reviewer 2 Report

I consider that all the requests were fulfilled. Therefore, I recommend publishing the manuscript in the present form.